# DNA Methylation-Specific Analysis of G Protein-Coupled Receptor-Related Genes in Pan-Cancer

**DOI:** 10.3390/genes13071213

**Published:** 2022-07-07

**Authors:** Mengyan Zhang, Jiyun Zhao, Huili Dong, Wenhui Xue, Jie Xing, Ting Liu, Xiuwen Yu, Yue Gu, Baoqing Sun, Haibo Lu, Yan Zhang

**Affiliations:** 1Computational Biology Research Center, School of Life Science and Technology, Harbin Institute of Technology, Harbin 150001, China; myzhang_bioinfo@163.com (M.Z.); zjy18845768427@163.com (J.Z.); 20s028049@stu.hit.edu.cn (H.D.); xuewenhui0118@126.com (W.X.); xjaixuexi@163.com (J.X.); gycarrie@hit.edu.cn (Y.G.); 2College of pathology, Qiqihar Medical University, Qiqihar 161042, China; liuting3406@163.com (T.L.); yxw05@sohu.com (X.Y.); 3State Key Laboratory of Respiratory Disease, Guangzhou Medical University, Guangzhou 510089, China; sunbaoqing@vip.163.com; 4Department of Gastrointestinal Medical Oncology, Harbin Medical University Cancer Hospital, Harbin 150000, China

**Keywords:** G protein-coupled receptor, DNA methylation, biomarker, drug sensitivity

## Abstract

Tumor heterogeneity presents challenges for personalized diagnosis and treatment of cancer. The identification method of cancer-specific biomarkers has important applications for the diagnosis and treatment of cancer types. In this study, we analyzed the pan-cancer DNA methylation data from TCGA and GEO, and proposed a computational method to quantify the degree of specificity based on the level of DNA methylation of G protein-coupled receptor-related genes (GPCRs-related genes) and to identify specific GPCRs DNA methylation biomarkers (GRSDMs) in pan-cancer. Then, a ridge regression-based method was used to discover potential drugs through predicting the drug sensitivities of cancer samples. Finally, we predicted and verified 8 GRSDMs in adrenocortical carcinoma (ACC), rectum adenocarcinoma (READ), uveal Melanoma (UVM), thyroid carcinoma (THCA), and predicted 4 GRSDMs (F2RL3, DGKB, GRK5, PIK3R6) which were sensitive to 12 potential drugs. Our research provided a novel approach for the personalized diagnosis of cancer and informed individualized treatment decisions.

## 1. Introduction

The precise diagnosis of cancer can reduce cancer mortality [1]. Diagnostic methods of cancer have been developed with a variety of biomarkers. Many studies have shown and verified the diagnosis of cancer in some biomarkers [2,3]. Zhan et al. obtained 34 signatures which could distinguish HCC cases from healthy subjects with high sensitivity (80.4%) and specificity (79.4%) [4]. Pu et al. identified and validated a panel of prostate cancer-specific DNA methylation biomarkers with diagnosis value in silico [5]. The mining of these biomarkers could diagnose a certain type of cancer more accurately. However, there are extensive heterogeneity between cancers; therefore, the development of biomarkers between pan-cancer can reduce certain detection steps and costs, and can also make it easier to guide clinical diagnosis and prognosis.

G protein-coupled receptors (GPCRs) involve controlling initiation and progression of tumors [6]. GPCRs are the largest family of cell membrane receptors, and nearly half of the molecular drugs in clinical medicine target GPCR signals to treat various diseases of the central nervous system, cardiovascular system and metabolic system [7,8]. It has been suggested that highly expressed GPCRs in cancer cells (e.g., GPRC5A in PDAC and colon cancer cells and GPR68 in PDAC CAF) may contribute to malignant phenotypes and may be novel therapeutic targets for cancer as biomarkers [9]. In addition, it has been reported that some GPCRs are either over- or underexpressed in tumor tissue compared with adjacent healthy tissue in different cancers. Sriram et al. analyzed The Cancer Genome Atlas (TCGA) data for mRNA expression, mutations, and copy number variation (CNV) in 20 categories and 45 subtypes of solid tumors and quantified differential expression of GPCRs by comparing tumors against normal tissue from the Gene Tissue Expression Project (GTEx) database. They found GPCRs were overrepresented among coding genes with elevated expression in solid tumors and GPCRs may be used as a biomarker for cancer [10]. Many studies have shown extensive methylation of GPCR in cancer, such as differential methylation of G protein-coupled receptor signaling genes in gastrointestinal neuroendocrine tumors [11].

Although the research on the drug target and prognosis biomarkers of GPCRs in cancer has gradually deepened, the biology relevance of GPCRS for the malignant phenotype means that GPCRS may be a contribution that has not been fully explored in cancer. The application of GPCRs-related genes as cancer-specific signatures would be promising.

The apparent genetic mechanism helps the occurrence and development of cancer, and the epigenetic mutation promotes the dynamic gene expression mode that promotes the evolution of tumor and adapts [12]. DNA methylation is tissue-specific and plays a decisive role in tissue differentiation [13], and has been widely used as a biomarker in cancer diagnosis and treatment due to its stability. Zou et al. applied the quantitative characteristics of DNA methylation to study the prognosis of cancer [14]. DNA methylation is confirmed to participate in the stage of cancer earlier [15], and DNA methylation is related to normal development and growth and is dysregulated in tumors [16,17]. DNA methylation in promoter region is closely related to tumorigenesis and studies have proved that methylation of promoter influence diagnosis and treatment [18,19]. Additionally, the DNA methylation levels of GPCRs-related genes will reveal a specific biological difference in pan-cancer and become reliable biomarkers in the earlier stage.

In this study, we developed a specific quantitative method to identify the specific biomarkers in pan-cancer based on DNA methylation of GPCRs-related genes. The contributions are summarized as follows: (1) a novel approach to identify specific biomarkers for pan-cancer; (2) unlike previous work, we obtained GRSDMs for pan-cancer, which ensured specific biomarkers only diagnose a cancer type and avoided redundant testing steps and costs; (3) the specific biomarkers for classify cancer types had a superior performance, achieving an average AUC of 0.88 on the testing datasets; (4) we further predicted the drug targets of these specific biomarkers in pan-cancer to personalized treatment.

In conclusion, specific biomarkers (GRSDMs) in pan-cancer could satisfy convenient and stable diagnostic criteria, preventing unnecessary tests. This study illustrates that a computational method was developed to quantify the degree of specificity based on the level of DNA methylation of GPCRs-related genes and identified GRSDMs in pan-cancer, providing prospects for early diagnosis and treatment in pan-cancer.

## 2. Materials and Methods

### 2.1. Dataset and Data Preparation

We downloaded DNA methylation data of 33 cancer types generated with the Illumina Infinium HumanMethylation450 BeadChip array from the TCGA data portal, and survival data of the corresponding samples, as well as the total of 11 cancer and normal DNA methylation data from the GEO database, filtered out some cancer types without normal samples, and finally DNA methylation data of the 26 cancer types for further analysis from TCGA database (detailed in Appendix A). The genomic transcription initiation site upstream of 2000 and 500 bp downstream were used as the promoter region, and then we focused the gene methylation data on the promoter region. The data was preprocessed to remove methylation sites with missing β values exceeding 70% of the total number of samples, and the KNN method was used to supplement the null values to further standardize the data. The “sva” R package was applied to bridge the batch effects between TCGA and GEO. The IC50 values of 198 drugs were downloaded from GDSC database.

### 2.2. Identify Differentially Methylated Sites and Genes

The DNA methylation cases of each cancer type from TCGA were de-batched with the corresponding cancer type data from GEO, in order to ensure that the β values of the obtained DNA methylation sites conformed to a uniform distribution. We randomly divided each type of cancer samples into 7:3 as a training set and a testing set. We applied *t*-test method to filter the differentially methylated sites of cancer samples and normal tissue samples on the training set in every cancer type, obtaining a set of *p* values and FDR (*p* < 0.05 and FDR < 0.05), and then mapped the differential methylation sites to genes based on platform annotation information and the differentially methylated genes were visualized with pheatmap package version 1.0.8 (Tartu, Estonia; https://cran.r-project.org/web/packages/pheatmap/index.html, accessed on 4 January 2019 ) in R 4.1.1.

### 2.3. Functional Annotation of Differentially Methylated Genes

We used the “ClusterProfiler” R package [20] to annotate differentially methylated genes on The Kyoto Encyclopedia of Genes and Genomes (KEGG) pathways. Significantly enriched pathways were defined as FDR < 0.05.

### 2.4. Feature Selection of Differentially DNA Methylated Sites

The differential DNA methylation sites of cancer samples and adjacent tissue samples of each cancer type were selected using the “Boruta” algorithm to obtain the characteristic DNA methylation sites of cancer types for distinguishing cancer from non-cancer samples. The different characteristic DNA methylation sites sets were used to build a decision tree classifier to distinguish between different cancer types and normal samples, respectively. Area under curve (AUC) was used to evaluate the performance of the models.

### 2.5. Identification of Pan-Cancer-Specific GPCRs-Related DNA Methylation Genes (GRSDMs)

A union of characteristic DNA methylation genes by “Boruta” for each cancer type as characteristic DNA methylation genes. The average value of the characteristic DNA methylation genes in each type of cancer $m_{i,N}$ was used as the β value of the characteristic DNA methylation genes, which was input into QDMR to obtain the entropy $H_{i}$ of each characteristic DNA methylation gene. $p_{i,N}$ was the relative gene i methylation probability between cancers N. S represented the number of the samples in each cancer.
(1)$$m_{i,N}=\frac{\sum_{s=1}^{s}m_{i}}{S}$$
(2)$$p_{i,N}=m_{i,N}/\sum_{N=1}^{N}m_{i,N}$$
(3)$$H_{i}=-\sum_{N=1}^{N}p_{i,N}\log_{2}(p_{i,N})$$

In order to eliminate the heterogeneity between the internal samples of the cancer, we also calculated the average value of all genes in each cancer $\bar{m_{i,N}}$ and the coefficient of variation of gene in each cancer type $C_{i,N}$. $C_{i,N}$ was the gene standard deviation $SD_{i,N}$ of the average DNA methylation level of gene $m_{i,N}$ in all cancer types. The product of $C_{i,N}$ and $\bar{m_{i,N}}$ as a heterogeneous indicator in each cancer type samples. Cancer gene-specific values $T_{i,N}$ were calculated.
(4)$$\bar{m_{i,N}}=\frac{\sum_{i=1}^{i}m_{i,N}}{i}$$
(5)$$C_{i,N}=\frac{SD_{i,N}}{\bar{m_{i,N}}}\times 100\%$$
(6)$$T_{i,N}=\frac{H_{i}\times m_{i,N}}{C_{i,N}\times \bar{m_{i,N}}}$$

We constructed the dissociation formula of cancer gene specificity value to qualitatively measure the specificity of cancer and specific DNA methylation genes, which made each specific DNA methylation gene correspond to only one cancer type. The average of the specific values of each gene corresponding to all cancers was $\bar{X_{i}}$. $T_{i,N}(\max)$ and $T_{i,N}(\min)$ were the max value and the min value, respectively, in the specific values of each gene corresponding to all cancers.
(7)$$\bar{X_{i}}=\frac{\sum_{N=1}^{N}T_{i,N}}{N}$$
(8)$$M_{i,N}=\left|\frac{T_{i,N}-\bar{X_{i}}}{T_{i,N}(\max)-T_{i,N}(\min)}\right|$$

Finally, $M_{i,N}$ was used as an indicator to identify the specific DNA methylation gene signature GRSDMs of cancers, and the calculated maximum value of each gene corresponding to all cancers indicated the specific DNA methylation gene signature of cancer.

### 2.6. Survival Analysis

Pan-cancer-specific DNA methylation genes for survival analysis based on progression free survival (PFS), and *p* value < 0.05 was used as the threshold to analyze the prognostic genes. Kaplan-Meier (KM) curves were drawn for independent factors related to survival.

### 2.7. Correlation Analysis with Drug Targets

We downloaded IC50 values of 198 drugs from GDSC database. We used “oncoPredict” R package [21] to predict IC50 in every samples from TCGA based on mRNA gene expression data. Spearman correlations were applied to calculate the correlations of drug IC50 values with validated DNA methylation levels of GRSDMs separately, in order to find potential effective drug treatments for GRSDMs.

## 3. Results

### 3.1. The Landscape of DNA Methylation Genes Related with GPCRs in Pan-Cancer

To assess whether the DNA methylation levels of GPCRs-related genes can be signatures of specific DNA methylation genes in pan-cancer, we analyzed the DNA methylation levels of all GPCRs-related genes. An overview and Kruskal–Wallis test were performed to examine the differences in DNA methylation levels of GPCRs-related genes in pan-cancer. At the pan-cancer level, the DNA methylation levels of GPCRs-related genes showed significant difference (*p* < 0.05), For instance, ACC was significant different from UVM, but some cancers did not show significant differences due to noise from other GPCRs-related genes. BRCA and CESC did not show difference. Screening effective DNA methylation biomarkers based on the GPCRs was a crucial way to eliminate noise and better diagnose cancer types. (Figure 1).

### 3.2. Acquisition of Characteristic DNA Methylation Genes in Pan-Cancer

After matching DNA methylation cancer and normal samples, cancer types without normal samples were deleted, and finally 26 types of cancer were analyzed. In order to obtain GRSDMs in pan-cancer, we first analyzed the differential DNA methylation sites for each type of cancer in training set, and obtained a total of 26 differential DNA methylation sites and genes sets (Figure 2A, Appendix A). These differential DNA methylation genes performed functional enrichment, and significantly enriched in cAMP signaling pathway and Rap1 pathway and so on (Figure 2B). The number of differential DNA methylation sites/genes for every type of cancer was detailed in Appendix A. The characteristic differential DNA methylation sites/genes were screened by “Boruta” method, and finally, 26 characteristic DNA methylation sites/genes of pan-cancer were obtained to distinguish cancer from non-cancer. For example, in UVM, we obtained 485 DNA methylation sites; there were 153 characteristic DNA methylation sites in ACC (Figure 2C). The characteristic DNA methylation genes in pan-cancer were defined as union set of every characteristic DNA methylation genes in each cancer type. The 3283 characteristic DNA methylation genes on the training set and 1382 GPCRs-related genes were intersected to obtain 169 GPCRs-related characteristic DNA methylation genes in pan-cancer (Figure 2D) and then were used to construct a decision tree classifier to test the classification performance of the characteristic DNA methylation sites, and the AUCs of the classification model for up to 25 cancer types such as ACC reached 1.0 on the testing set (Figure 2E, Appendix A).

### 3.3. Identifying Pan-Cancer-Specific GPCRs-Related DNA Methylation Genes (GRSDMs)

We calculated the average value of DNA methylation levels of genes in each cancer type, and obtained the entropy value of 169 genes after the calculation of QDMR (Appendix A). Then, 169 characteristic DNA methylation GPCRs-related genes were filtered by the pan-cancer-specific DNA methylation identification algorithm (Methods), and each gene had its numerical value indicating the degree of specificity. Finally, each gene was specific to a certain cancer. Three cancers had only one specific DNA methylation gene, such as breast cancer (BRCA), cervical squamous cell carcinoma and endocervical adenocarcinoma (CESC), stomach adenocarcinoma (STAD). Four cancers had more than 10 specific DNA methylation genes identified, such as colon adenocarcinoma (COAD), THCA, thymoma (THYM), and UVM (Figure 3A). The specific degree values of the DNA methylation genes in pan-cancer was detailed in Appendix A. To better classify cancer types, we used 169 GRSDMs to construct a random forest to diagnose cancer, and the AUCs of the model could reach 0.88 on testing set and 0.829 on GEO dataset, respectively (Figure 3B,C).

### 3.4. Prognostic Analysis of Specific DNA Methylation Genes

To screen out specific DNA methylation genes associated with prognosis, we used the mean β value of 169 specific DNA methylation genes for each cancer as the threshold for high and low risk groups. Finally, we obtained 22 specific DNA methylation genes that make the survival curves of the high and low risk groups significantly separated (*p* < 0.05) (Figure 4, Appendix A).

### 3.5. Validation of Specific DNA Methylation Genes in GEO

In order to verify reliable GRSDMs, 22 specific DNA methylation genes related to prognosis obtained in the training set were verified on 11 sets of GEO data, and 8 of the specific DNA methylation genes (DGRK, F2RL3, GRK5, NECAB2, OR1C1, OR2L13, OR6F1, and PIK3R6) had been verified in pan-cancer. For example, gene DGKB of UVM in TCGA is specifically high, and gene DGKB was also specifically high in the GEO data set or the testing dataset; DNA methylation levels of gene F2RL3 was significant higher in ACC than other cancer types in the training, testing and GEO validation sets. The eight GRSDMs all displayed consistency in three datasets. (*p* < 0.0001) (Figure 5).

### 3.6. Selection of Potential Drugs Based on Specific DNA Methylation Genes in Pan-Cancer

Since GRSDMs can explain the heterogeneity of pan-cancer, the relative drug sensitivity was analyzed based on specific DNA methylation genes to select potential drugs for different cancer types and improve prognosis. We performed a spearman correlation analysis between 8 validated specific DNA methylation genes levels in 4 cancer types and the IC50 values of 198 drugs downloaded from GDSC, and we finally screened 4 GRSDMs that were significant correlation with 12 related drugs. In ACC, the sensitivities of specific DNA methylation gene F2RL3 and drug BI.2536, sepantronium bromide, were significantly negatively correlated; in UVM, DNA methylation levels of gene DGKB, GRK5 and PIK3R6 were negatively correlation with 10 kinds of drugs (cor < −0.4, *p* < 0.05). Specifically, F2RL3 was negatively correlated with BI.2536 in training and testing sets in ACC. NECAB2 and PIK3R6 were correlated with trametinib or sepantronium bromide in training and testing sets in UVM, which may predict that these drugs had effects on the hypermethylation of specific DNA methylation genes in ACC and UVM. (Figure 6). When hypermethylation of the gene was detected in a patient, the use of the drug may have a positive therapeutic effect.

## 4. Discussion

GPCR dysregulation is related to a variety of human diseases and disorder [22]. Novel GPCRs that are changed in cancer have been discovered in genome-wide comprehensive investigations of different human malignancies. Ding et al. found genes of frequently hypomethylated DMCs (hypomethylated in at least eight cancers) are mainly enriched in GPCRs [23]. At the same time, the identification of the diagnosis of cancer is of great significance for the treatment of cancers, which can reduce unnecessary examinations. DNA methylation status is closely associated with diverse diseases, and is generally more stable than gene expression, which might be useful in supporting clinical decisions [24]. Many studies reflected DNA methylation genes were used as biomarkers for diagnosis and investigated the DNA methylation profile of human pan-cancer [23,25]. Ding et al. identified seven CpG sites that could effectively discriminate tumor samples from adjacent normal tissue samples for 12 main cancers of TCGA with AUC of 0.99. However, these studies mostly analyzed the DNA methylation profiles of pan-cancer, and the “multiple-to-many” cancer gene markers will increase unnecessary expenses and waste the effective time of early treatment. In the study, we applied GPCRs-related genes to the recognition of specific labels of pan-cancer, which will reliably guide the early diagnosis and treatment of cancer.

We first found that the DNA methylization level of the GPCRs-related genes were widespread significant in pan-cancer. However, the differences in some cancers were not obvious, which may be the noise from some GPCRs-related genes. Therefore, we identified the specific biomarkers of cancers based on the DNA methylization level of GPCRs-related genes. We finally predicted and verified the total of eight special DNA methylated signs of four cancer types. Due to the insufficient DNA methylation data of cancers, some of the specific biomarkers we recognized had not been completely verified. Another reason was that the integration of datasets from GEO database would still have noise, and some biomarkers did not fully meet the standards we verified. The eight biomarkers showed significant correlations with the IC50 values of four kinds of drugs. Among them, F2RL3 methylation as specific biomarker in ACC and sepantronium bromide and BI.2536 were sensitive to it. F2RL3 methylation in blood is a strong predictor of mortality [26]. Through the blood detection of F2RL3, the diagnosis of cancer and screening related drugs may be diagnosed early. BI.2536 could be a potential drug for treatment of ACC, and it was validated to treat in diabetic kidney disease [27]. We also predicted some potential drugs in Uveal Melanoma (UVM), such as selumetinib and trametinib. Clinically, selumetinib compared with chemotherapy resulted in a modestly improved progression-free survival and response rate of patients with advanced UVM [28]. Neoadjuvant plus adjuvant dabrafenib and trametinib significantly improved event-free survival versus standard of care in patients with high-risk, surgically resectable, clinical stage III-IV melanoma [29,30]. Identifying specific markers and finding potentially reliable drugs was important for personalized treatment of different cancers.

Our research provided a direction for the identification method of the specific biomarkers of the pan-cancer and insights on the treatment of drugs for related cancer. The types of cancer in this study had not been widely studied, due to the shortcomings of normal samples, and more comprehensive cancer type data could better explore the corresponding specific biomarkers. There was another limitation. The cancer data used by the GEO database for verification was not enough. If the cancer type verification data were sufficient, the recognition of our specific biomarkers may be more comprehensive and stable.

## 5. Conclusions

We developed a method of quantitative DNA methylation-specific degrees to obtain specific GPCRs-related DNA methylation genes in pan-cancer. Finally, we obtained and validated eight specific DNA methylation genes to diagnose four cancer types, and we predicted four kinds of drugs that may have sensitivity to eight biomarkers. Our research showed that identifying cancer-specific biomarkers was important for early diagnosis, and it could provide new directions for drug treatment.

## Figures and Tables

**Figure 1 genes-13-01213-f001:**
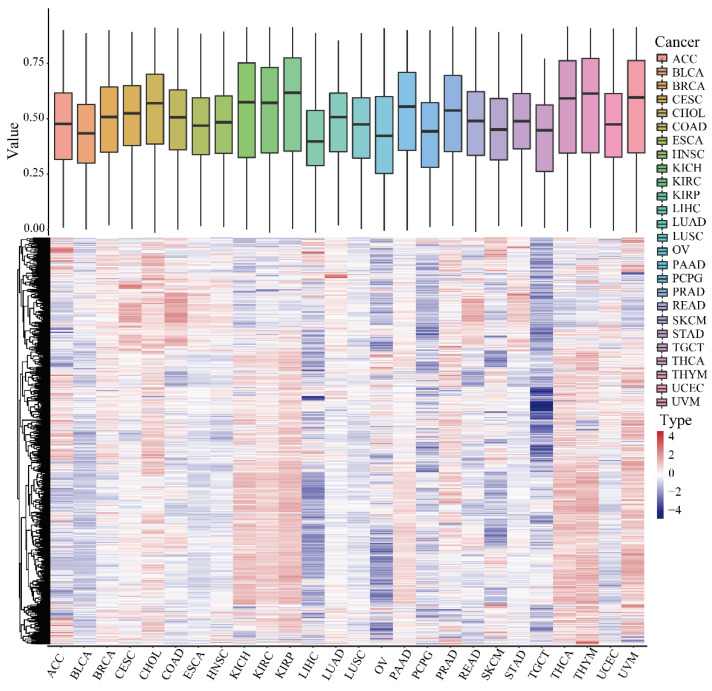
The overview of DNA methylation of G protein-coupled receptors in pan-cancer. The bars in the upper half represents the distribution of DNA methylation values of all samples in each cancer, and the heatmap in the lower half shows the DNA methylation levels of GPCRs-related genes in different cancer types.

**Figure 2 genes-13-01213-f002:**
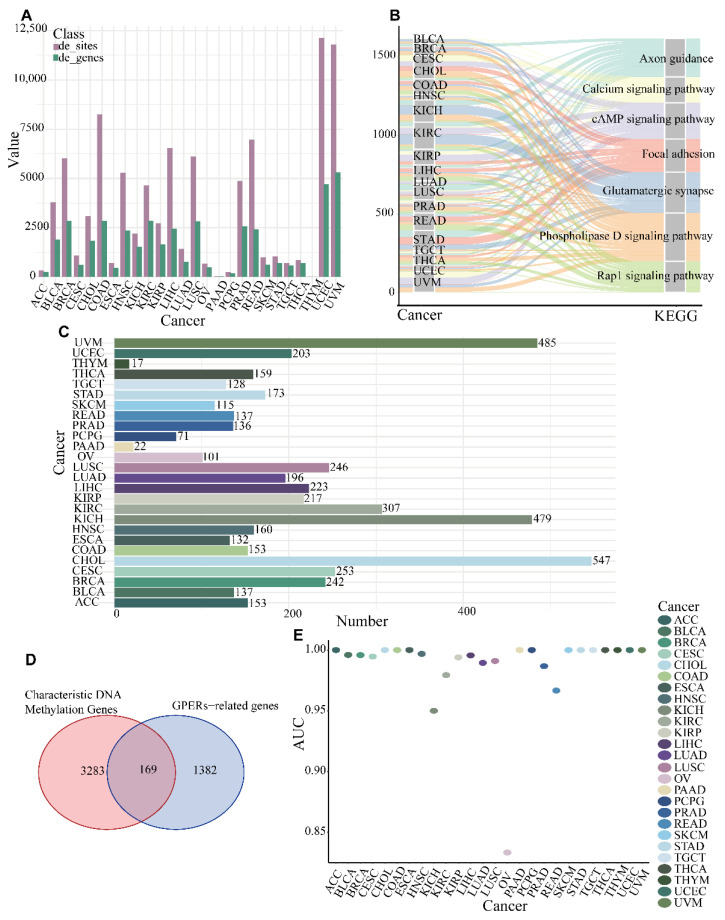
The acquisition of characteristic DNA methylation genes. (**A**) The number of differential DNA methylation sites/genes; (**B**) KEGG pathway of differential gene enrichment; (**C**) The number of characteristic DNA methylation genes by “Boruta” method in pan-cancer; (**D**) The intersection of characteristic DNA methylation genes and GPERs-related genes; (**E**) The AUCs of classification for cancer/non-cancer in pan-cancer.

**Figure 3 genes-13-01213-f003:**
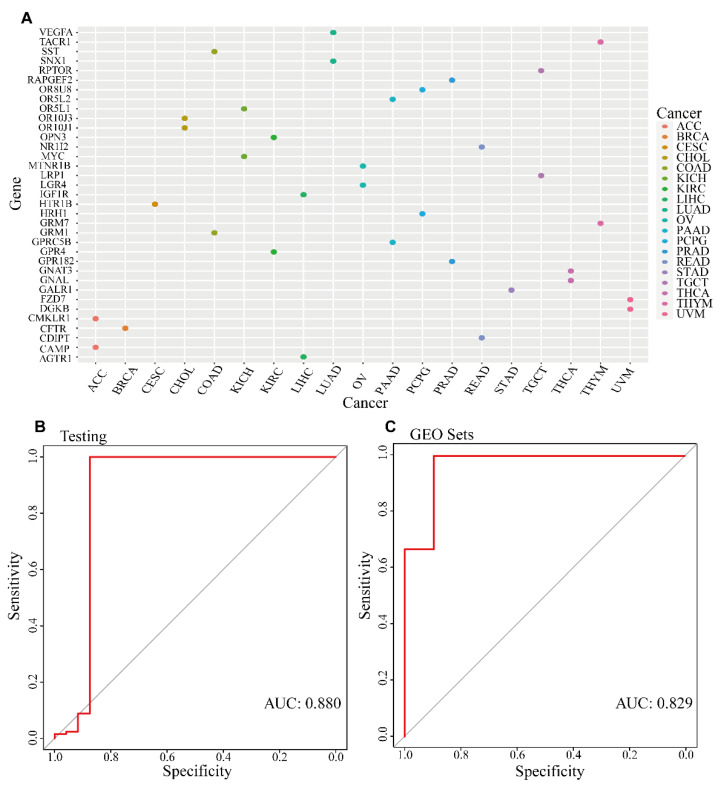
GRSDMs recognition and model establishment. (**A**) GRSDMs in pan-cancer; (**B**) Random forest model performance in the test set; (**C**) Random forest model performance in the GEO validation set.

**Figure 4 genes-13-01213-f004:**
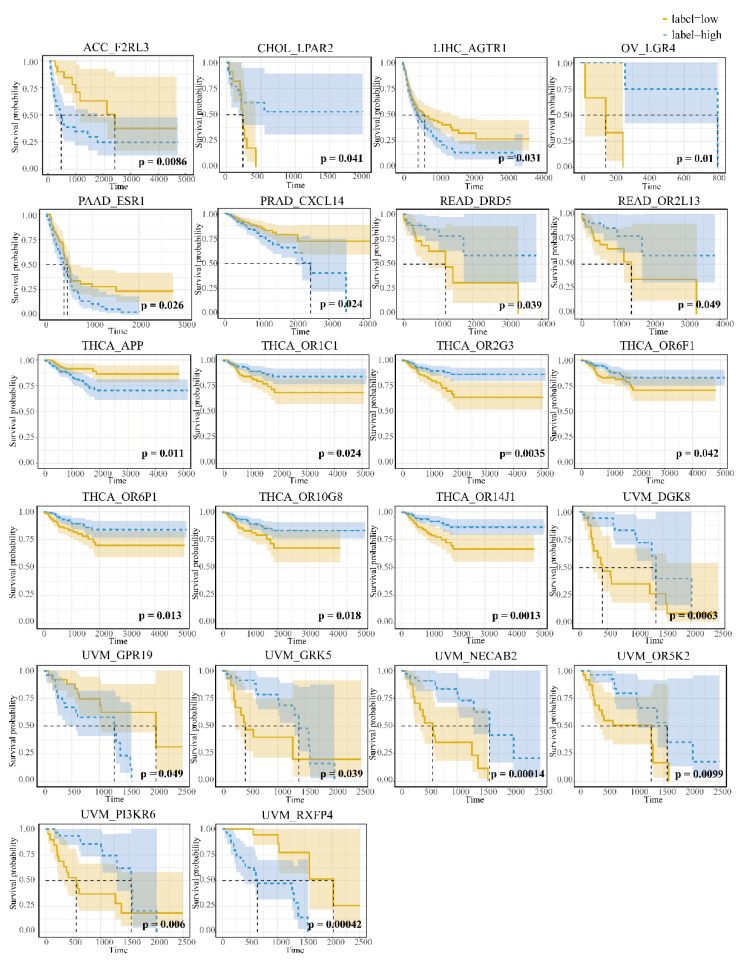
The survival curves of GRSDMs with significant related prognosis.

**Figure 5 genes-13-01213-f005:**
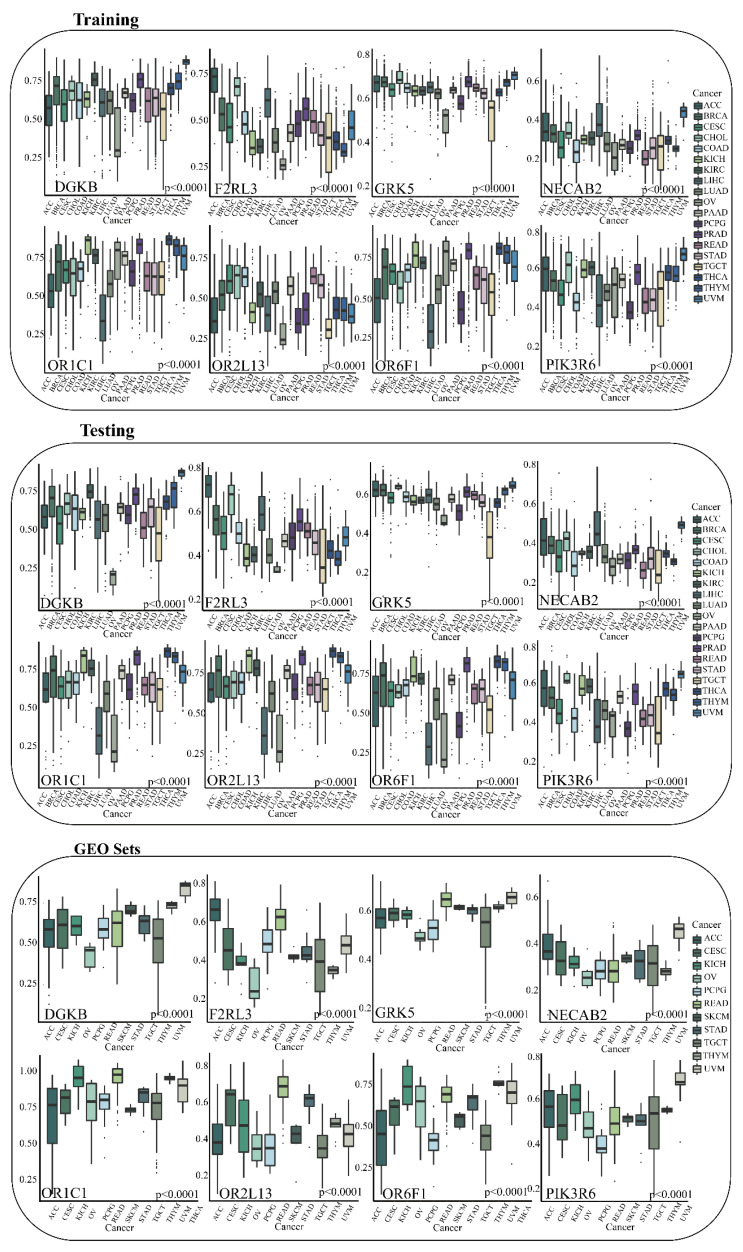
The validation of GRSDMs in training, testing and GEO validation sets.

**Figure 6 genes-13-01213-f006:**
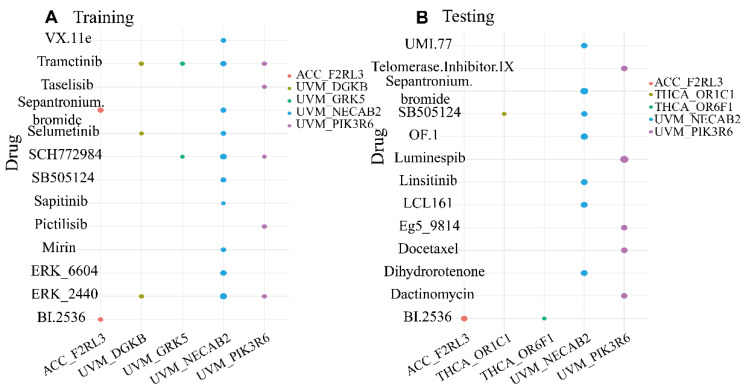
Significantly correlated GRSDMs and IC50 of drugs. (**A**) IC50s of drugs for significantly correlated GRSDMs in the training set. (**B**) IC50s of drugs for significantly correlated GRSDMs in the testing set.

## Data Availability

All authors declare that all data supporting the findings of this study are available in the article. Data and code are available in the following github repository: https://github.com/HIT-CBC/GPCRs_pancancer.git (accesed on 1 June 2022).

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
