# Peer review of "DNA Methylation-Specific Analysis of G Protein-Coupled Receptor-Related Genes in Pan-Cancer"

_genes, 2022, doi:10.3390/genes13071213_

Round 1

Reviewer 1 Report

The manuscript “DNA methylation-specific analysis of G protein-coupled receptor-related genes in pan-cancer” by Zhang et al., demonstrates novel results of bioinformatic analysis of Illumina 450K data, including TCGA data and GEO datasets. The Authors presented a specific quantitative method to identify the specific biomarkers in pan-cancer based on DNA methylation of G protein-coupled receptor-related genes (GPCRs).

The Authors decided to search for cancer biomarkers within the GPCRs – please describe the GPCRs and provide a comprehensive explanation on choosing those genes as potential candidates for cancer biomarkers. The Authors have just stated that “G protein-coupled receptor (GPCRs) involves controlling initiation and progression of tumors [6]” and described some studies indicating that “some GPCRs are either over- or underexpressed in tumor tissue compared with adjacent healthy tissue (or normal) in different cancers” (pages 1-2). G protein-coupled receptors (GPCRs) comprise the largest family of receptor proteins in mammals and play important roles in many physiological and pathological processes. Gene expression of GPCRs is temporally and spatially regulated, and many splicing variants are also described. Therefore, the Authors should be more careful with the conclusion that “The application of GPCRs-related genes as cancer-specific signatures will be relatively stable and effective biomarkers.”

Moreover, the Authors decided to analyze the DNA methylation data for GPCRs from Illumina 450K platform, using TCGA data and GEO datasets – it is also not clear why? – are the DNA methylation alterations of GPCRs-related genes common in different cancers? They have analyzed only the promoter region of the GPCRs – why only the promoter region? The methylation changes (hypo- or hypermethylation) within the gene body are common as well in cancer cells.

The Authors “downloaded DNA methylation data of 33 cancer types generated with the Illumina 450K array from the TCGA data portal, and survival data of the corresponding samples, as well as the total of 11 cancer and normal DNA methylation data from the GEO database” – how many samples have they downloaded for each cancer type (n = ?) and what GEO numbers they have used in the analysis? Have the Authors downloaded and analyzed only DNA methylation data for tumor tissue and adjacent healthy tissue or have they used also DNA methylation data for blood samples? Looking for DNA methylation biomarkers in blood from cancer patients instead of hardly accessible tumor tissue would be more reasonable.

All the Figures (1-6), especially the captions on the Figures are hard to read.

Round 2

Reviewer 1 Report

The authors have addressed all of my comments and concerns in the revised version. I have no additional comments.

Reviewer 2 Report

In my opinion, the revised munuscript is good.